# The Role of Aquaporin Overexpression in the Modulation of Transcription of Heavy Metal Transporters under Cadmium Treatment in Poplar

**DOI:** 10.3390/plants10010054

**Published:** 2020-12-29

**Authors:** Andrea Neri, Silvia Traversari, Andrea Andreucci, Alessandra Francini, Luca Sebastiani

**Affiliations:** 1BioLabs, Institute of Life Sciences, Scuola Superiore Sant’Anna, Piazza Martiri della Libertà 33, 56127 Pisa, Italy; andrea.neri.sr@gmail.com (A.N.); sil.traversari@gmail.com (S.T.); luca.sebastiani@santannapisa.it (L.S.); 2Department of Biology, University of Pisa, via Luca Ghini 13, 56126 Pisa, Italy

**Keywords:** aquaporin, cadmium, metal excess, metal transport, mineral elements, *Populus alba*

## Abstract

*Populus alba* ‘Villafranca’ clone is well-known for its tolerance to cadmium (Cd). To determine the mechanisms of Cd tolerance of this species, wild-type (wt) plants were compared with transgenic plants over-expressing an aquaporin (*aqua1*, GenBank GQ918138). Plants were maintained in hydroponic conditions with Hoagland’s solution and treated with 10 µM of Cd, renewed every 5 d. The transcription levels of heavy metal transporter genes (*PaHMA2*, *PaNRAMP1.3*, *PaNRAMP2*, *PaNRAMP3.1*, *PaNRAMP3.2*, *PaABCC9*, and *PaABCC13*) were analyzed at 1, 7, and 60 d of treatment. Cd application did not induce visible toxicity symptoms in wt and *aqua1* plants even after 2 months of treatment confirming the high tolerance of this poplar species to Cd. Most of the analyzed genes showed in wt plants a quick response in transcription at 1 d of treatment and an adaptation at 60 d. On the contrary, a lower transcriptional response was observed in *aqua1* plants in concomitance with a higher Cd concentration in medial leaves. Moreover, *PaHMA2* showed at 1 d an opposite trend within organs since it was up-regulated in root and stem of wt plants and in leaves of *aqua1* plants. In summary, *aqua1* overexpression in poplar improved Cd translocation suggesting a lower Cd sensitivity of *aqua1* plants. This different response might be due to a different transcription of *PaNRAMP3* genes that were more transcribed in wt line because of the importance of this gene in Cd compartmentalization.

## 1. Introduction

Cadmium (Cd) is a toxic element that can affect mineral homeostasis, photosynthesis, water balance, and nutrient uptake in plants [1,2,3]. The physiological modifications and the dynamics of Cd accumulation and distribution within plant tissues as well as the function of the main heavy metal transporters involved in Cd uptake, translocation, and detoxification are well-known in poplar plants [4,5,6,7,8,9].

After the mobilization from soil, Cd uptake within plant tissues is determined by plasma membrane transporters located in root cells. When Cd enters in root tissues, translocation to shoot depends on its mobility within the root symplast, across the endodermal barrier, and further within the xylem. Cd accumulation and storage in leaf cells require its uptake from xylem and apoplast, as well as its phloem loading, to mobilize and allocate it among leaves or other organs (i.e., seeds, fruits) [10]. Several putative metal transporter genes are involved in these processes, such as transporters belonging to: (i) Zinc/Iron-regulated transporter-like Proteins (ZIPs), (ii) Natural Resistance and Macrophage Proteins (NRAMPs), and (iii) plasma membrane Heavy Metal proton-ATPases (HMAs) [10,11]. Members of NRAMP family are considered general metal ion transporters due to their ability to transport several ions, including Cd [12,13]. These transporters play a crucial role in the response, translocation, and accumulation of Cd [14,15,16].

On the other hand, Cd long-distance transport toward the shoot is primarily driven by the xylem water flux; when inside the stele, the divalent cation is actively translocated to the xylem vessels through transporters belonging to the HMA family [17,18]. In *Populus trichocarpa* the *HMA* isoforms *PtHMA1*–*PtHMA4* belong to the Zn/Co/Cd/Pb subgroup while *PtHMA5*–*PtHMA8* are members of the Cu/Ag subgroup [19]. In several species, the *HMA1*–*4* isoforms are crucial for Cd translocation and accumulation [20,21,22,23]. Eren and Argüello [24] demonstrated that in *Arabidopsis thaliana*, HMA2 is an active pump which drives Zinc (Zn) and Cd efflux to the extracellular compartment. Metal compartmentalization in plant cell is essential because free Cd within the cytosol causes phytotoxicity and the activation of specific processes of detoxification [25]. Several ATP Binding Cassette (ABC) transporters have vacuolar localization and, in particular, *AtABCC3/6/7* are involved in Cd active storage for protection [26]. *AtABCC1* and *AtABCC2* genes play an important role in metal compartmentalization as well [27], and an increase in transcript levels of *ABCC10* was found in Cd-exposed roots of *P. trichocarpa* and *Populus deltoides* [28]. Recently, a study found that overexpression of the *ABCG36* gene of *Populus tomentosa* is effective in enhancing Cd tolerance in Arabidopsis plants, indicating the function of this transporter as a heavy metal extrusion pump [29].

In addition to transporter activity, Cd movement from the root to aerial tissues is driven by the water flow within the xylem [1]. It has been demonstrated that Cd uptake in a plant induces a decrease in tissue water content [30,31,32,33]. In particular, Cd can affect the conductivity of aquaporin proteins (AQPs) leading to a decrease in membrane water permeability [34]. Moreover, the overexpression of aquaporin genes increased Cd resistance in tobacco plants [35].

AQPs belong to the super family of membrane proteins known as major intrinsic protein (MIP) family. This family of channel proteins is divided in four subgroups: (i) Plasma membrane Intrinsic Proteins (PIPs), (ii) Tonoplast Intrinsic Proteins (TIPs), (iii) Nodulin 26-like Intrinsic Proteins (NIPs), (iv) Small basic Intrinsic Proteins (SIPs) [36,37]. MIPs permit the movement of water or small neutral solutes, as well as metalloids such as arsenic, in a wide variety of organisms [38].

Actually, 55 *AQP* genes have been identified in *P. trichocarpa* [39]. Di Baccio et al. [40] reported the transcriptome analyses of *Populus* × *euramericana* ‘I-214’ clone leaves and found a down-regulation of an aquaporin (*aqua1*, GenBank GQ918138) under Zn excess. The *aqua1* down-regulation has been found in the roots as well, using an RNA sequencing approach in plants maintained in the same growth conditions [41]. Ariani et al. [42] tested the heavy metal tolerant species *P. alba* ‘Villafranca’ clone overexpressing the *aqua1* gene under Zn excess. Authors reported that, under Zn treatment, an increase in the relative growth rate and the intrinsic transpiration efficiency of this modified line compared to the wild-type (wt) plants occurs. However, *aqua1* overexpression did not increase Zn accumulation even if it gave higher tolerance to root tissues [42]. Furthermore, Ariani et al. [43] demonstrated that AQUA1 is a Hg-sensitive aquaporin regulated at both transcriptional and post-translational levels in response to Zn treatment that determines the re-localization of AQUA1 in the new forming vesicles. However, *aqua1* line has never been studied under Cd treatment. For these reasons, in this study plants of *P. alba* ‘Villafranca’ clone wt and overexpressing the *aqua1* gene (*aqua1*) were exposed to Cd treatment. The aim was to understand if overexpression of the *aqua1* gene in poplar: (i) enhanced short- and long-term Cd transport and accumulation; (ii) interfered with the gene transcription of metal transporters related to Cd uptake, translocation, and accumulation.

## 2. Materials and Methods

### 2.1. Plant Growth and Cd Treatment

Two lines of *Populus alba* (L.) ‘Villafranca’ clone, wt and over-expressing the aquaporin *aqua1* (69 times more than the wt in leaves, as reported by Ariani et al. [42]), were first grown in vitro with half-strength woody plant medium (WPM) [44] and then transferred to pots filled with perlite (Laterlite, Milano, Italy) inside Plexiglas boxes with 100% relative humidity. The AQUA1:GFP line over-expressing *aqua1* revealed a strong localization of this gene in roots and leaf guard cells on in vitro plantlets [42].

During the acclimation process, the nutrient solution was gradually changed from WPM half-strength liquid medium to Hoagland’s solution at pH 6.2 [45] and the humidity was reduced. After the acclimation period in growth chamber (23:18 °C day:night temperature, 65–70% relative humidity, and 16 h photoperiod at photosynthetic photon flux density of 400 μmol m^−2^ s^−1^ supplied by fluorescent lights), wt and *aqua1* plants (*n* = 9) were transferred into plastic pots, containing 8–20 Ø mm expanded ATAMI clay (Atami B.V., Rosmalen, The Netherlands) and maintained in hydroponic conditions with Hoagland’s solution, renewed every 5 d, and continuously aerated by aquarium pumps (250 l h^−1^). Plants were treated without Cd (0 μM Cd, Control) or with Cd (10 μM Cd) supplied as cadmium nitrate hexahydrate (Cd(NO_3_)_2_·6H_2_O, Sigma-Aldrich, St. Louis, MO, USA), and iron as Fe-tartrate instead of Fe-EDTA, to avoid Cd chelation. Plants were sampled at 1, 7, and 60 d from the beginning of the Cd treatments (*n* = 3). Relative Growth Rate (RGR) was determined as the difference between the natural logarithm of dry weight (DW) at 60 d and the natural logarithm of the mean DW at the start of the experiment, divided for the day of treatment (60 d).

### 2.2. Photosystem II Efficiency

Photosystem II (PSII) efficiency was measured on dark-adapted (30 min) leaves (Leaf Plastochrone Index, LPI, between 10 and 13, following the procedure reported by Erickson and Michelini [46]) using a portable chlorophyll fluorometer (FMS 2, Hansatech, Inc., Norfolk, UK). Dark-adapted leaves were exposed to an excitation light intensity of 3000 μmol m^−2^ s^−1^ (600 W m^−2^) emitted by a halogen light source. Background fluorescence signal (F_0_) and the maximum fluorescence (F_m_) were measured and the maximum quantum efficiency of photosystem II was determined as:(1)$$F_{V}/F_{m}=\left(F_{m}-F_{0}\right)/F_{m}$$

The non-photochemical quenching (NPQ) was determined as well. Finally, electron transport rate (ETR) was calculated as:(2)ETR=PAR×0.5×ΦPSII×0.84
where ΦPSII is the quantum yield of electron transfer of PSII.

### 2.3. Cadmium, Manganese, and Zinc Analyses

Plants were harvested after 1 d, rinsed with deionized water, and separated into leaves, stem, and root. After 7 and 60 d, leaves were designated as apical (1 ≤ LPI ≤ 6), medial (7 ≤ LPI ≤ 18), and basal (LPI > 18) on the basis of LPI. Roots were washed with 10 mM CaCl_2_ to remove mineral elements adsorbed to the root surface. Plant organ samples were dried in a forced-circulation oven at 65 °C and ground with a laboratory mill (IKA-Werke GmbH & Co.KG, Staufen, Germany). Ground samples (0.2 g) were digested with 5 mL HNO_3_ followed by 1 mL of HClO_4_. The resulting solution was analyzed using an atomic absorption spectrometer (AAnalyst 200, Perkin-Elmer, Waltham, MA, USA). Two analytical reference standards for Cd, Zn, and manganese (Mn) were used as control (WEPAL IPE, Wageningen University): *Daucus carota* (L.) leaf (307 ± 71 µg kg^−1^ of Cd, 25.0 ± 2.93 mg kg^−1^ of Zn and 42.1 ± 4.53 mg kg^−1^ of Mn, certified concentrations) and shoot (3070 ± 491 µg kg^−1^ of Cd, 185.0 ± 16.4 mg kg^−1^ of Zn and 427.0 ± 40.6 mg kg^−1^ of Mn, certified concentrations).

### 2.4. Gene Identification

*A. thaliana* gene sequences were retrieved in the TAIR database (www.arabidopsis.org, [47]). The following sequences were selected: AT4G30110 (*AtHMA2*), AT1G30400 (*AtABCC13*), AT3G60160 (*AtABCC9*), AT1G80830 (*AtNRAMP1*), AT1G47240 (*AtNRAMP2*), and AT2G23150 (*AtNRAMP3*). The amino acidic sequences were used as seeds for a Blastp analysis in *P. trichocarpa* proteome on the Phytozome database, using the matrix BLOSUM62 (https://phytozome.jgi.doe.gov). The sequences showing significant hits were downloaded from the database and primers were designed using Primer3 program (http://primer3.ut.ee/) and used for RT-PCR analyses. The primer sequences are reported in Appendix A. A summary of all transporters investigated in this study is reported in Table 1.

### 2.5. RNA Extraction and RT-PCR Analyses

Fresh frozen samples (200 mg) were ground with liquid nitrogen and total RNA extracted using “Spectrum™ Plant Total RNA Kit” and “On-Column *DNase* I Digest Set” (Sigma-Aldrich, St. Louis, MO, USA), using the centrifuge Allegra 64R (Beckman) according to manufacturer’s procedure. RNA concentration was quantified with a “SPECTRO StarNano” spectrometer (BMG, Labtech GmbH, Ortenberg, Germany), while RNA integrity was checked by an electrophoresis gel (1% *w*/*v* agar, Tris-Acetate-EDTA buffer). The RNA was transcribed to cDNA using an “iScript cDNA Synthesis Kit” (BIORAD, Hercules, CA, USA) according to manufacturer’s procedure. cDNA was used in RT-PCRs to determine target gene mRNA accumulations, compared with *18S* housekeeping gene transcription, using the ΔΔCq method [52]. RT-PCRs were performed by “Eco Real Time PCR System” (Illumina, CA, USA) using a “x HOT FIREPol^®^ EvaGreen^®^ qPCR Supermix” kit (Solis Biodyne, Tartu, Estonia) according to manufacturer’s procedure. At the end of each RT-PCR, the melting curve was evaluated to exclude secondary amplification products.

### 2.6. Statistical Analyses

Two-way ANOVA analyses were performed on data of photosystem II efficiency, RT-PCRs, and mineral element concentrations. Means were subjected to a Tukey’s multiple comparison test (*p* < 0.05). When two-way ANOVA was not significant for the interaction between the variables Cd and line and the Tukey’s post hoc test was not performed, the differences in gene transcription between control and treated plants were assessed with a *t*-test (*p* < 0.05).

Statistical analyses were performed with NCSS 2000 Statistical Analysis System Software. All graphs were plotted with Prism 9 software (GraphPad, La Jolla, CA, USA). The heat-maps were elaborated using RStudio software (Boston, MA, USA). Detailed ANOVA results are reported in Appendix A.

## 3. Results

### 3.1. Physiological Parameters and Mineral Element Concentrations

After 60 d of Cd exposure, RGR did not change among lines and treatments (Table 2) and plants treated with Cd did not show any visible chlorosis symptom (data not shown). The evaluation of photosystem II efficiency confirmed that after 1 and 7 d both treated lines were not different from the control plants (Table 2). Only after 60 d of Cd treatment, both *aqua1* and wt plants showed a significant increment in NPQ in comparison to the control plants (+29.4% and 19.6%, respectively).

Cd concentrations in plant organs increased progressively from 1 until 60 d in both adapted lines (Figure 1). Overall, roots displayed the highest Cd concentrations after 60 d (905 μg g^−1^ DW in wt, 841 μg g^−1^ DW in *aqua1*, *p* = ns). Cd increased in roots and stem immediately at 1 d of treatment while it was below the detection limit within the shoot. At 7 d of treatment, Cd was retrieved within the apical leaves of both lines (16.8 ± 4.88 Cd µg g^−1^ DW and 19.0 ± 1.41 Cd µg g^−1^ DW in wt and *aqua1*, respectively, *p* = ns), within the medial leaves of *aqua1* plants, while it was not detected in the basal leaves of both lines. In detail, at 7 d the Cd concentration in medial leaves of *aqua1* plants was higher than in wt plants (9.45 ± 0.17 μg g^−1^ DW vs. 1.31 ± 0.43 μg g^−1^ DW, respectively, *p* < 0.001) and it remained higher even at 60 d of treatment (85.8 ± 10.08 μg g^−1^ DW vs. 47.5 ± 12.37 μg g^−1^ DW, respectively, *p* = 0.004). At 60 d, Cd increased also within the basal leaves of both wt and *aqua1* plants.

Zn and Mn concentrations were analyzed to detect possible alterations in mineral element uptake due to the Cd treatment. Zn concentration was not affected by Cd treatment in all the organs, except for apical leaves at 7 d where it was lower than in control plants (Appendix A). Mn was more influenced by Cd treatment and starting from 7 d it decreased within root, stem, and basal leaves of treated plants and it remained lower even at 60 d (Figure 2).

### 3.2. Gene Transcription

The selected genes were identified in *P. trichocarpa* database and the Arabidopsis homologue sequences were retrieved in TAIR database as follows: Potri.001G044900 homologue to *AtNRAMP1;* Potri.002G121000 homologue to *AtNRAMP2*; Potri.007G050600 and Potri.007G050700 homologues to *AtNRAMP3*; Potri.006G076900 homologue to *AtHMA2*; Potri.014G180100 homologue to *AtABCC9*; and Potri.004G034800 homologue to *AtABCC13.* For these genes the following nomenclature was used: *PaNRAMP1.3*, *PaNRAMP2*, *PaNRAMP3.1* and *PaNRAMP3.2*, *PaHMA2*, *PaABCC9*, and *PaABCC13*, respectively. All genes analyzed were hypothesized to be involved in the direct transport of Cd (*PaNRAMP1.3*, *PaNRAMP2*, *PaNRAMP3.1*, *PaNRAMP3.2*, and *PaHMA2*) or in the transport of metal-conjugates in the vacuole (*PaABCC9*, *PaABCC13*).

The heat map (Figure 3), allowed the identification of genes that were similarly influenced by Cd treatment within the lines. The most altered gene was *PaHMA2* in both lines that did not cluster with other genes and was localized as an outgroup. Indeed, *PaHMA2* resulted particularly altered by Cd excess, but very differently between wt and *aqua1* plants. The heat maps highlighted that under Cd treatments an up-regulation of the genes *PaNRAMP3.1*, *PaABCC13*, *PaNRAMP3.2*, *PaABCC9*, *PaNRAMP2*, occurred in wt plants at root and stem levels while in *aqua1* plants the same gene appeared up or down-regulated (Figure 3). In wt line, the *PaNRAMP3.1* and *PaNRAMP2* showed similar transcription patterns to *PaABCC13* and *PaABCC9*, respectively. On the contrary, in *aqua1* plants, *PaNRAMP3.1* gene clustered with *PaNRAMP1.3* while *PaNRAMP2* with *PaABCC13*. The transcript accumulation of the studied metal transporter genes resulted in being particularly altered at 1 d of Cd treatment in root and stem of wt plants (Appendix A) where a stronger up-regulation was reported in comparison with *aqua1* treated plants. Interestingly, few genes showed an opposite trend in wt and *aqua1* plants since the transgenic line had a down-regulation or no difference in transcription instead of an up-regulation under Cd excess, such as *PaABCC9*, *PaNRAMP2*, and *PaNRAMP3.2*

At 1 d of Cd treatment, *PaHMA2* was strongly up-regulated in root and stem of wt plants (Figure 4). In *aqua1* plants *PaHMA2* was slightly up-regulated in roots and, in particular, in leaves. At 7 d, this gene was up-regulated in stem and medial leaves of wt treated plants and up-regulated in apical leaves of both lines. At 60 d of Cd exposure, *PaHMA2* remained up-regulated only in medial leaves of both lines while its transcription was strongly suppressed in the other plant parts (Figure 4c).

At 1 d of Cd treatment, *PaNRAMP3.1* (Figure 5) was up-regulated in all organs of treated plants, particularly in wt roots (about seven-fold more transcribed than in *aqua1*). However, at 7 d *PaNRAMP3.1* was still up-regulated only in wt stem of treated plants (Figure 6) while at 60 d it was up-regulated in root, stem, and apical leaves of both wt and *aqua1* treated plants and it was up-regulated in basal leaves of *aqua1* treated plants (Figure 7). At 1 d of treatment, *PaNRAMP3.2* had an opposite response within the leaves of the two treated lines (less transcribed in wt plants and more transcribed in *aqua1* plants), and it resulted strongly up-regulated in stem of wt treated plants and weakly up-regulated in root of *aqua1* treated plants (Figure 6). At 7 d it resulted as up-regulated only in stem of wt treated plants (Figure 6) while it had not statistically significant differences at 60 d (Figure 7).

*PaNRAMP1.3* was up-regulated at 7 d in stem of wt and in apical leaves of *aqua1* treated plants (Figure 6), while it was up-regulated in stem of both wt and *aqua1* plants at 60 d, mostly in wt plants in which it was about six-fold more transcribed (Figure 7).

Regarding *PaNRAMP2*, a strong up-regulation was observed at 1 d in stem of treated wt plants while this gene was up-regulated at 1 d in leaves and at 7 d in root and in apical leaves of *aqua1* treated plants (Figure 5 and Figure 6).

Finally, the *PaABCC9* gene showed an up-regulation in root and leaves of wt plants after 1 d of Cd exposure (Figure 5), in particular in the roots (about five-fold more transcribed than in control plants), while an up-regulation occurred in stem of wt plants at 7 d and in stem of both line at 60 d (Figure 6 and Figure 7). The *PaABCC13* was up-regulated in root and stem of wt treated plants at 1 d (Figure 5) while it was down-regulated in the wt leaves. At 7 d, this gene was down-regulated in basal leaves of wt treated plants and up-regulated in root of *aqua1* treated plants (Figure 6). Stem and apical leaves showed an up-regulation also for the *PaABCC13* gene in both lines at 60 d of treatment (Figure 7).

## 4. Discussion

Plant genetic background regulates homeostasis of metals, such as Cd, and their accumulation within different plant organs. The roots play a crucial role in heavy metal accumulation since they are involved in the uptake and selection of the solutes that can be transported toward aerial tissues [53,54]. In our experimental conditions, Cd concentration immediately increased within the roots of both lines after 1 d of treatment and the heavy metal started to be translocated toward the stem but was undetected in leaves. This distribution agreed with Cd pattern previously reported in *P. alba* ‘Villafranca’ clone [8]. In the herbaceous plants *Brassica juncea* and *Thalapsi caerulescens,* Cd concentration increased within the shoots after 10 h of treatment [55], while in *Oryza sativa* after 6 h of treatment [56]. In poplars the delay in Cd uptake in leaves might be due to stem buffering or the different plant anatomy which did not facilitate the Cd movement toward the leaves. At 7 d of Cd treatment, both wt and *aqua1* plants started to accumulate Cd within the apical leaves. Cd and other metals can move in bidirectionally through the phloem [57] and be translocated from the root to the shoot through the xylem [58]. Salt et al. [55] reported that Cd is accumulated first in the leaves on the top of shoot in *B. juncea* and *T. caerulescens,* and then in the basal leaves. At 7 d, this mechanism seemed to be present also in poplar and, interestingly, the Cd concentration in medial leaves was significantly higher in *aqua1* plants. This result suggested that overexpression of the *aqua1* gene in poplar enhanced Cd transport and accumulation in the aerial tissues. This behavior might be due to a different functionality of hydraulic system between wt and *aqua1* plants, that allowed a different Cd movement. Ariani et al. [42] reported that *aqua1* plants under Zn stress had no difference in Zn concentration compared to the wt plants. On the contrary, in our experiment, Cd treatment led to a different Cd concentration in *aqua1* plants and differently influenced Mn uptake.

Considering the range of toxicity of Cd, around 8–12 μg g^−1^ [59], the high concentration (up to 85.8 μg g^−1^ DW) of Cd found in *P. alba* ‘Villafranca’ leaves further confirmed the high tolerance of this clone to Cd [7]. The capability to translocate Cd (200–400 μg g^−1^ DW of Cd) to aerial parts of several poplar clones has been also reported by Zacchini et al. [60] in hydroponic conditions. After 60 d of Cd exposure, although wt and *aqua1* leaves did not show any symptom of toxicity in terms of chlorosis, necrosis, and F_v_/F_m_ ratio, an indication of plant response to Cd stress was provided by NPQ increase. The NPQ is a key protective process for thermally dissipating the excess of light energy that plants employ to prevent the over reduction of PSII. A noticeable difference in this parameter was observed between *aqua1* and wt plants at 60 d suggesting that the whole-chain electron transport was more affected by Cd exposure in *aqua1* leaves. In *aqua1* plants, Zn stress affected the F_v_/F_m_ ratio in the apical leaves after 35 d of treatment highlighting a higher level of stress in photosynthetic system in comparison to the Cd stress applied in the current research [42].

All the analyzed transporter genes were influenced by Cd treatment, despite their role in metal ion transport. Indeed, Cd can be transported through Fe, Zn, and Mn transporters such as the members of ZIP, NRAMP, and HMA families [10] and it substitutes metals in the enzyme active sites [49]. *AtHMA2* gene encodes a transporter involved in the root to shoot translocation of metals [61]; it has been demonstrated that in *A. thaliana* and *Triticum aestivum*, *HMA2* is localized in the plasma membrane of pericycle cells and vascular bundles, acting as an efflux pump that could also drive Cd within the stele, showing a really high affinity for this heavy metal [17,24,48,62]. In our experimental conditions, *PaHMA2* seemed to have a role in short-term mobilization of Cd (at 1 d), showing opposite transcription patterns within different organs between wt and *aqua1* treated plants. In wt plants, *PaHMA2* seemed to be involved in the translocation of Cd from the root to the shoot as suggested by its strong up-regulation in root and stem as a short-term response. In *aqua1* plants, *PaHMA2* up-regulation at 1 d within leaves brings forward the fastest Cd phloem translocation shown at 7 d of treatment. Indeed, the crucial role of this gene in Cd translocation is well-known, as reported in *A. thaliana* [62]. Moreover, this gene was down-regulated at 7 d in apical and basal leaves of both lines while it showed an up-regulation precisely in the medial leaves of wt plants where Cd concentration was lower than in *aqua1* plants. This up-regulation in wt plants could be responsible of a Cd efflux from the cells toward the extracellular space [24] allowing a reallocation of heavy metal among the plant, reducing symplastic transport.

The *NRAMP* genes are involved in the remobilization and uptake of divalent cations such as Cd [10]. *AtNRAMP3* is localized in vacuole tonoplast of vascular bundles and the stele. This gene is responsible for the mobilization of Fe, Mn, and Cd between vacuole and cytoplasm [14,63] and has been found highly expressed in hyperaccumulator plants such as *T. caerulescens* [64]. *PaNRAMP3.1* showed a several-fold change increase in gene transcription after only 1 d of treatment in all organs of wt plants but only a slight increase in root and leaves of *aqua1* plants. Moreover, *PaNRAMP3.2* showed a several-fold change increase in gene transcription in stem of wt plants at 1 d as well. The strong up-regulations of these genes in wt plants suggested a strong perception of Cd in this line not reported in *aqua1* plants. Indeed, other authors proposed that *AtNRAMP3* regulated the Cd sensitivity by mobilizing this metal from the vacuole to the cytoplasm [15]. The lower perception of Cd in *aqua1* plants could account for the lower *PaHMA2* transcription as well.

*AtNRAMP1* is principally involved in Mn uptake but can transport also Fe and Cd [14]. In *A. thaliana,* it is expressed in all plant organs and is localized on the plasma membrane [65,66]. Indeed, *PaNRAMP1.3* up-regulation in wt and *aqua1* treated plants at 60 d in stem and root could probably be linked with the low Mn concentration in plant tissues which induced a Mn mobilization. This behavior could indirectly induce an increase in Cd transport within root and stem cells. *AtNRAMP2* is a gene required for root growth in *A. thaliana* and it is localized in the trans-Golgi network [67]. It is principally involved in Mn intracellular distribution and it has a fundamental role in photosynthesis and cellular redox homeostasis [50]. In our experimental conditions, *PaNRAMP2* did not show a clear role in the response to the increasing Cd concentration. *AtABCC9* and *AtABCC13* genes, also known as *AtMRP9* and *AtMRP13*, are involved in the vacuole compartmentalization of glutathione conjugates [51,68]. *PaABCC9* and *PaABCC13* seemed to have a function in Cd compartmentalization within different organs during the short-term response mostly in wt plants. The differences between lines were probably related also to the diverse Cd sensitivity conferred by the different transcription of *PaNRAMP3.1* and *PaNRAMP3.2*. Generally, *PaABCC9* and *PaABCC13* showed a long-term response within the stem of both lines.

In conclusion, the gene transcription analyses showed a different behavior to all the analyzed transporters highlighting a different metal homeostasis of wt and *aqua1* lines. Most of the analyzed genes seemed crucial in the wt Cd short-term response, showing a quick reaction at 1 d of treatment and an adaptation at 60 d (acclimation phase). In *aqua1* plants, the overexpression induced a faster uptake and transport of Cd in the aerial part and a higher Cd concentration in medial leaves in concomitance with a lower activation of metal transporter genes. Therefore, a possible role of this aquaporin in Cd transport has been suggested. This different behavior should be also connected to a lower Cd sensitivity of *aqua1* plants due to a different modulation of transcription of heavy metal transporters and alterations in water balance. Further investigations are needed to understand the function of AQUA1 in the water balance of poplar plants which could clarify the improvement in Cd transport of *aqua1* plants. This work further confirmed the importance of exploring the use of transgenic plants with improved heavy metal tolerance for promoting sustainable phytoremediation practices [69].

## Figures and Tables

**Figure 1 plants-10-00054-f001:**
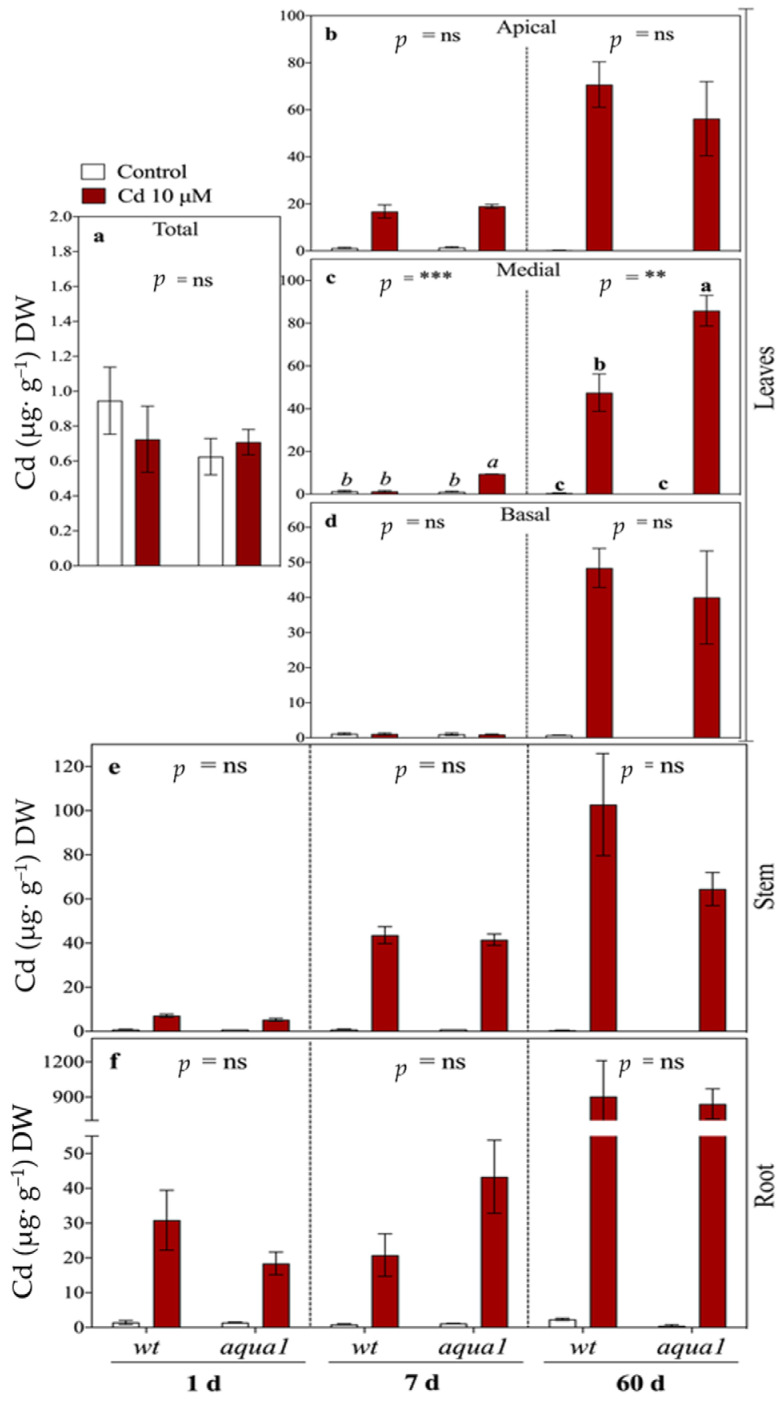
Cd concentrations (µg g^−1^ DW) in leaves (**a**–**d**), stem (**e**), and root (**f**). In the first sampling time (1 d) total leaves were analyzed (**a**) while at 7 and 60 d leaves were divided in three groups: apical (**b**), medial (**c**), and basal (**d**). Values represent the mean of three biological replicates ± SE. Data were analyzed with two-way ANOVA; *p*-values correspond to the interaction between Cd and Line are reported in the figure (** = 0.001 < *p* < 0.01, *** = *p* < 0.001, ns = not significant). When the interaction was significant, different letters indicate significant differences among treatments at each sampling time and organ. Cd and Line *p*-values are reported in Appendix A.

**Figure 2 plants-10-00054-f002:**
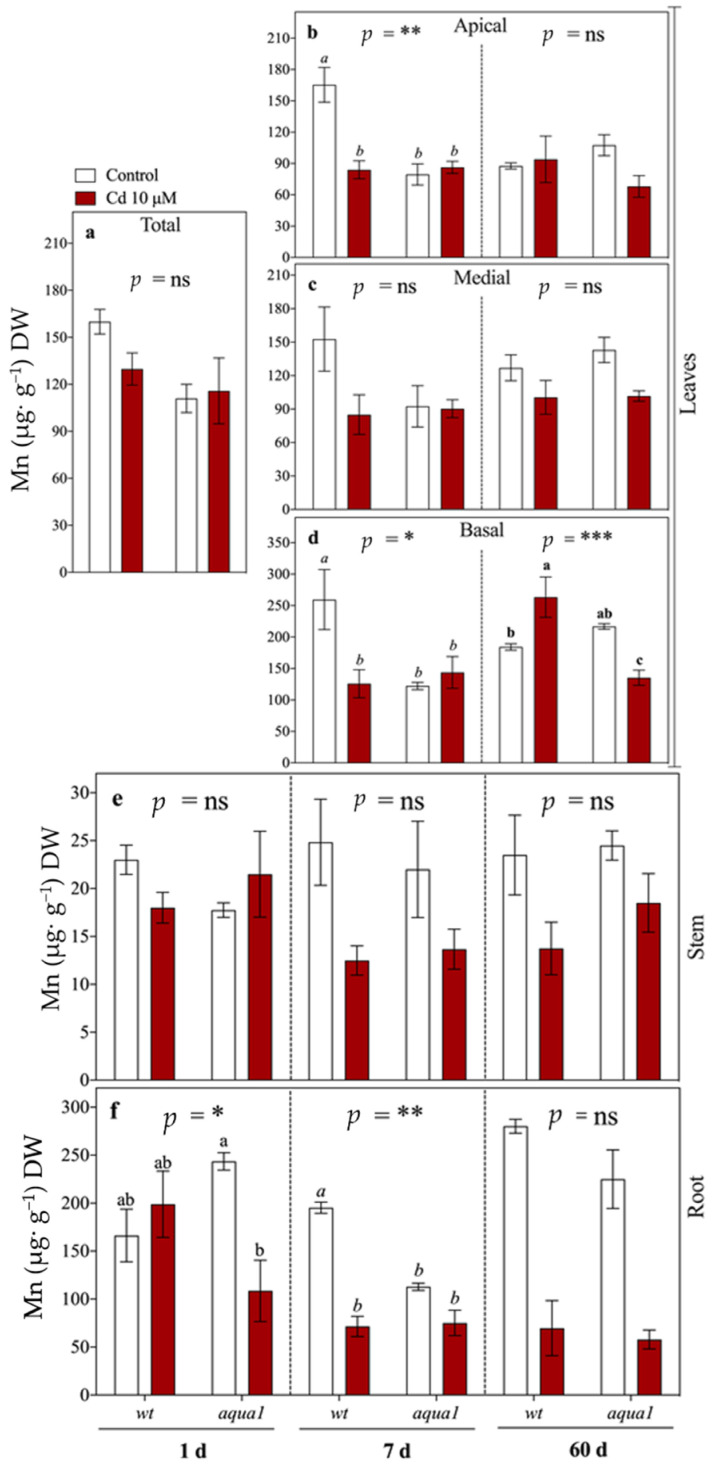
Mn concentrations (µg g^−1^ DW) in leaves (**a**–**d**), stem (**e**), and root (**f**) of wt and *aqua1* plants treated with 0 μM Cd (Control) or with Cd (10 μM Cd). In the first sampling time (1 d) total leaves were analyzed (**a**), while at 7 and 60 d leaves were divided in three groups: apical (**b**), medial (**c**), and basal (**d**). Values represent the mean of three biological replicates ± SE. Data were analyzed with two-way ANOVA; *p*-values correspond to the interaction between Cd and Line are reported in the figure (* = 0.01 < *p* < 0.05, ** = 0.001 < *p* < 0.01, *** = *p* < 0.001, ns = not significant). When the interaction was significant, different letters indicate significant differences among treatments at each sampling time and organ. Cd and Line *p* values are reported in Appendix A.

**Figure 3 plants-10-00054-f003:**
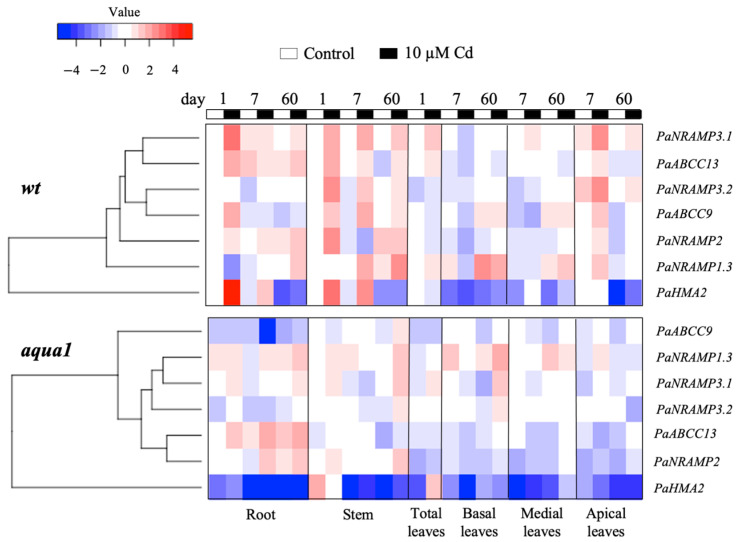
Heat maps of gene transcription levels in wt and *aqua1* plants at 1, 7, and 60 d in control and treated (10 µM Cd) conditions. Data displayed in the heat map are expressed in ln2^−ΔΔCt^.

**Figure 4 plants-10-00054-f004:**
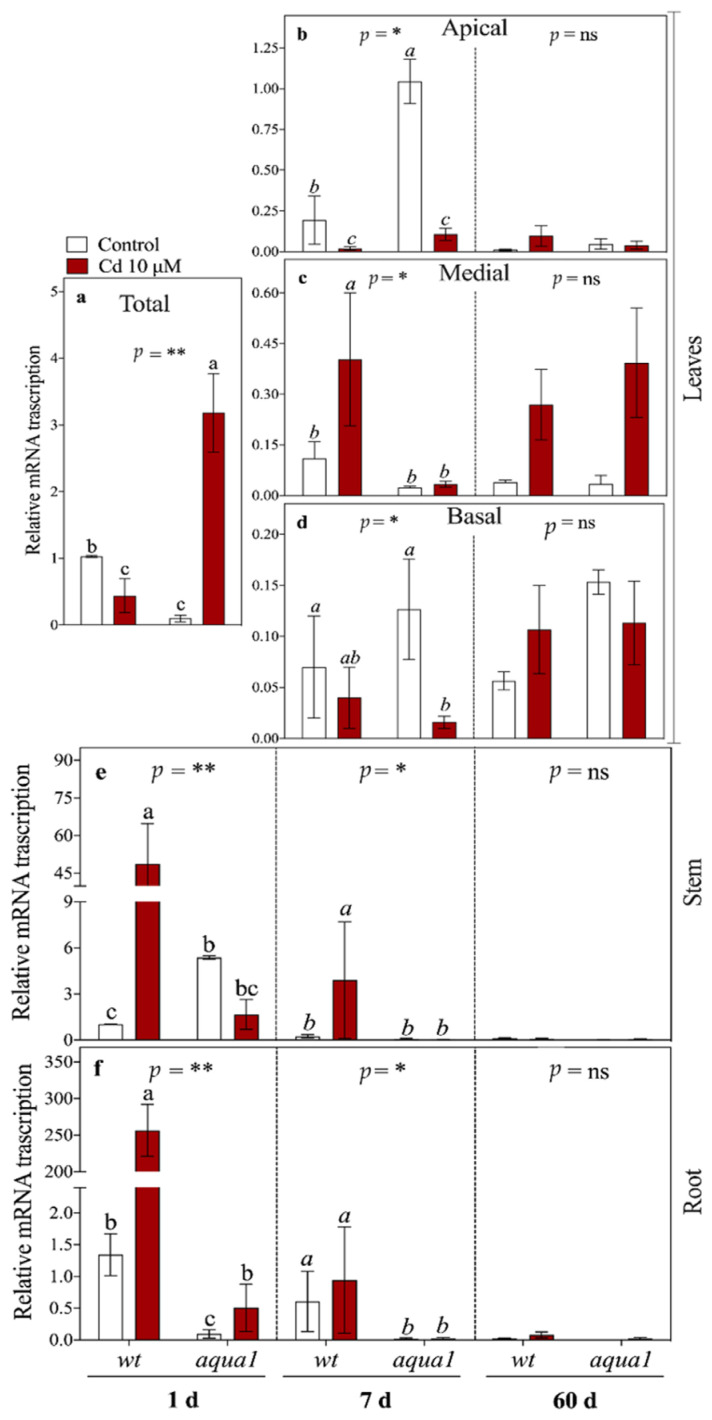
*PaHMA2* gene relative transcript abundances in leaves (**a**–**d**), stem (**e**), and root (**f**). In the first sampling time (1 d) the leaves were not divided in three groups (**a**) while at 7 and 60 d leaves were divided in three groups: apical (**b**), medial (**c**), and basal (**d**). Values represent the mean of three biological replicates ± SE. Data were analyzed with two-way ANOVA; *p*-values correspond to the interaction between Cd and Line are reported in the figure (* = 0.01 < *p* < 0.05, ** = 0.001 < *p* < 0.01, ns = not significant). When the interaction was significant, different letters indicate significant differences among treatments at each sampling time and organ. Cd and Line *p*-values are reported in Appendix A.

**Figure 5 plants-10-00054-f005:**
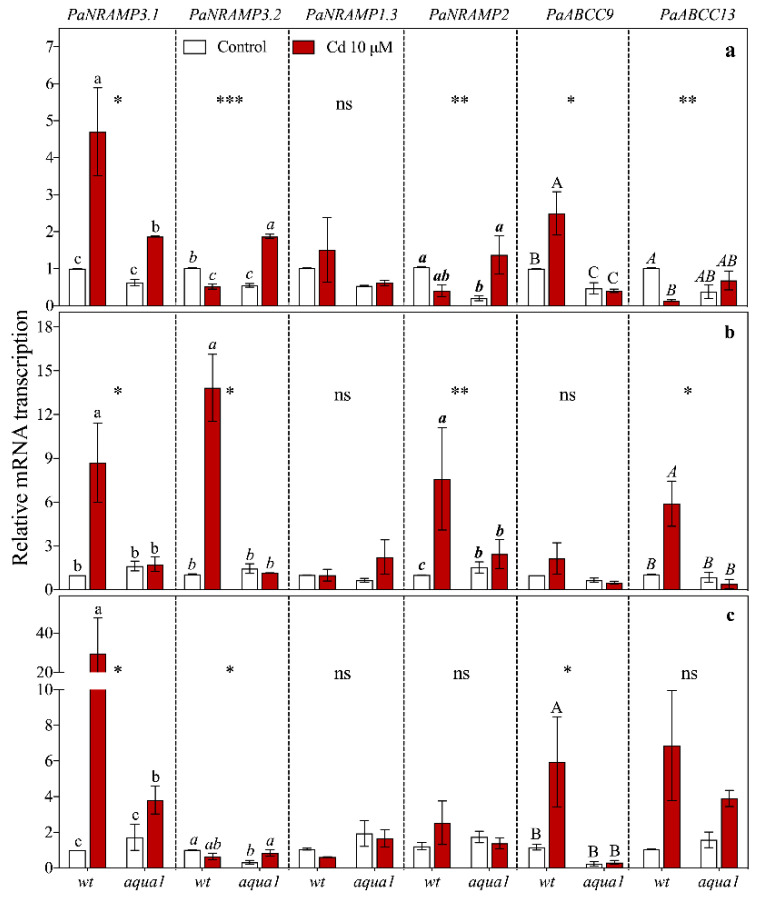
Gene relative transcript abundance in leaves (**a**), stem (**b**), and root (**c**) at 1 d of treatment. Values represent the mean of three biological replicates ± SE. Data were analyzed with two-way ANOVA; *p*-values correspond to the interaction between Cd and Line are reported in the figure (* = 0.01 < *p* < 0.05, ** = 0.001 < *p* < 0.01, *** = *p* < 0.001, ns = not significant). When the interaction was significant, different letters indicate significant differences among treatments at each sampling time and organ. Cd and Line *p*-values are reported in Appendix A.

**Figure 6 plants-10-00054-f006:**
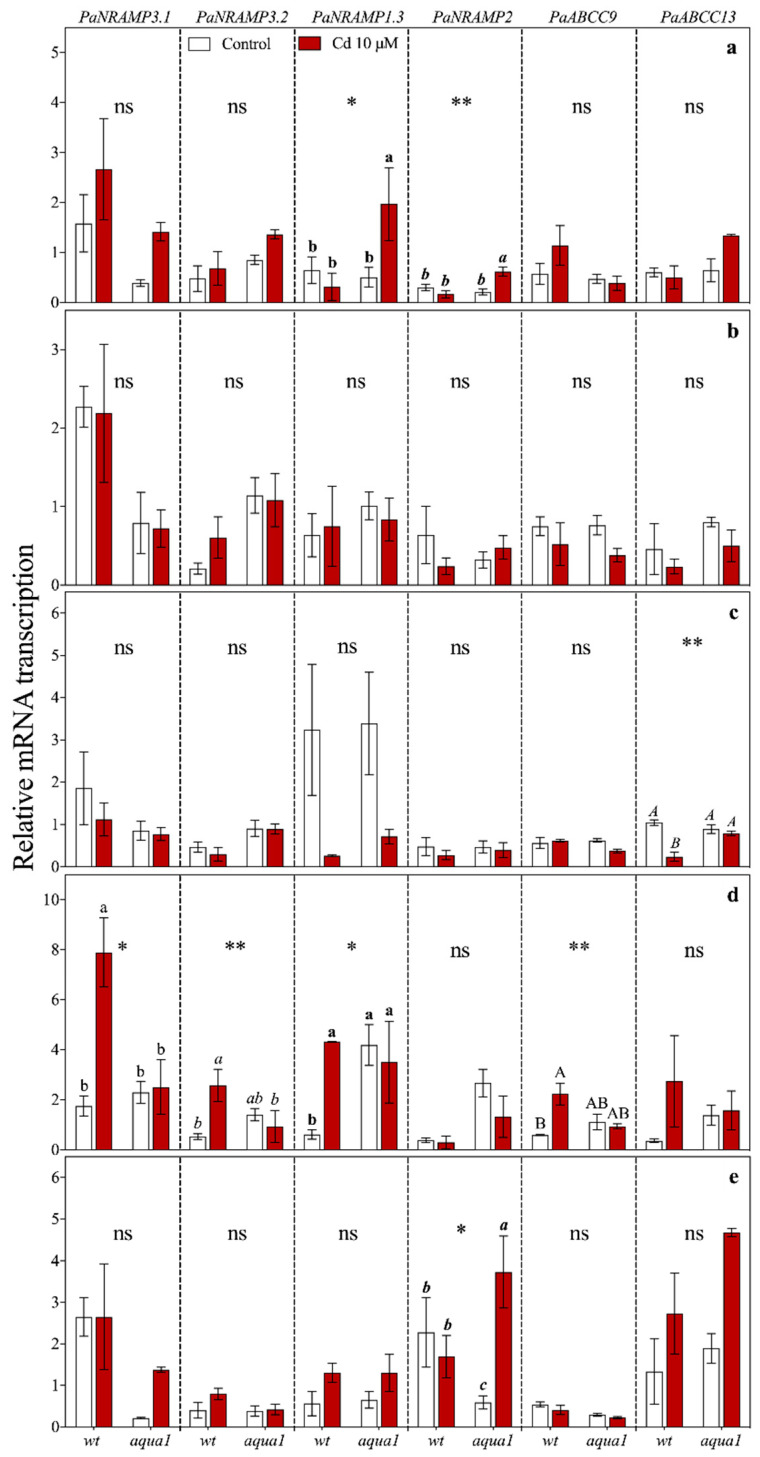
Gene relative transcript abundance in apical (**a**), medial (**b**), and basal (**c**) leaves, stem (**d**) and root (**e**) at 7 d of treatment. Values represent the mean of three biological replicates ± SE. Data were analyzed with two-way ANOVA; *p*-values correspond to the interaction between Cd and Line are reported in the figure (* = 0.01 < *p* < 0.05, ** = 0.001 < *p* < 0.01, ns = not significant). When the interaction was significant, different letters indicate significant differences among treatments at each sampling time and organ. Cd and Line *p*-values are reported in Appendix A.

**Figure 7 plants-10-00054-f007:**
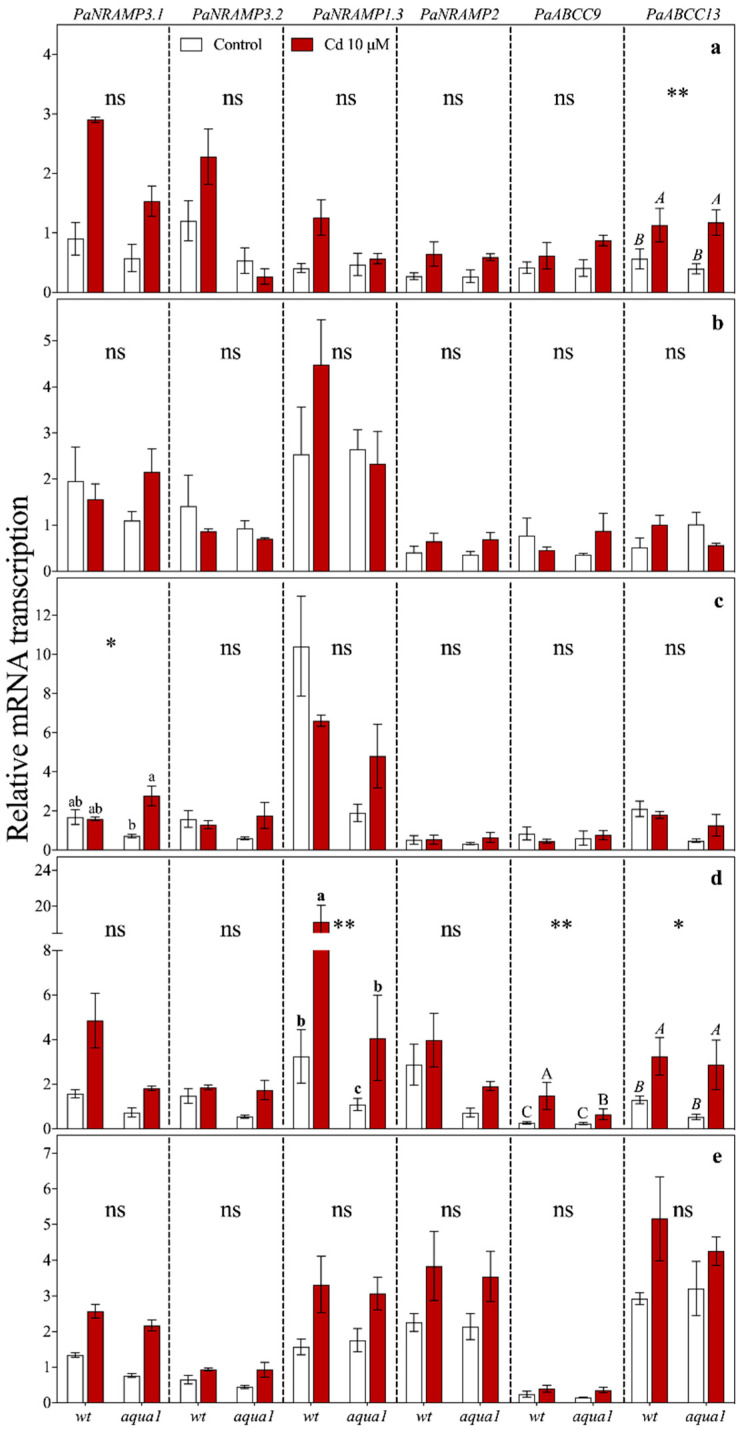
Gene relative transcript abundance in apical (**a**), medial (**b**), and basal (**c**) leaves, stem (**d**), and root (**e**) at 60 d of treatment. Values represent the mean of three biological replicates ± SE. Data were analysed with two-way ANOVA; *p*-values correspond to the interaction between Cd and Line are reported in the figure (* = 0.01 < *p* < 0.05, ** = 0.001 < *p* < 0.01, ns = not significant). When the interaction was significant, different letters indicate significant differences among treatments at each sampling time and organ. Cd and Line *p*-values are reported in Appendix A.

**Table 1 plants-10-00054-t001:** Main features of metal transporters investigated in this study in *Populus alba* plants.

Transporter	Metal Ions Transported	Species	Reference
NRAMP1.3	Mn, Fe	Arabidopsis	[13]
NRAMP2	Mn	Arabidopsis	[13]
NRAMP3	Fe, Mn	Arabidopsis	[13,48]
HMA2	Zn	Arabidopsis	[49]
ABCC9	Metal-Glutathione	Arabidopsis	[50]
ABCC13	Metal-Glutathione	Arabidopsis	[51]

**Table 2 plants-10-00054-t002:** Photosystem II efficiency parameters at 1, 7, and 60 d of Cd treatment and relative growth rate (RGR, g day^−1^), F_v_/F_m_ (maximum quantum efficiency of photosystem II), NPQ (non-photochemical quenching), ETR (electron transport rate) at 60 d calculated on dry biomass basis. The values represent the mean of three biological replicates ± SD. Data were analyzed with two-way ANOVA. The means were compared using a Tukey’s test (*p* < 0.05). Different letters indicate significant differences (*p* < 0.05). ns, not significant; * *p* < 0.05; ** *p* < 0.01; *** *p* < 0.001.

		Control		Cd 10 µM		ANOVA		
Day	Parameter	wt	*aqua1*	wt	*aqua1*	Cd	Line	Cd × Line
1	F_v_/F_m_	0.82 ± 0.010	0.82 ± 0.006	0.82 ± 0.007	0.82 ± 0.004	ns	ns	ns
	NPQ	0.19 ± 0.060	0.14 ± 0.051	0.19 ± 0.075	0.14 ± 0.104	ns	ns	ns
	ETR	3.5 ± 0.47	3.5 ± 0.54	3.4 ± 0.48	3.5 ± 0.92	ns	ns	ns
7	F_v_/F_m_	0.82 ± 0.001	0.82 ± 0.002	0.83 ± 0.004	0.83 ± 0.008	ns	ns	ns
	NPQ	0.14 ± 0.065	0.15 ± 0.063	0.14 ± 0.088	0.21 ± 0.013	ns	ns	ns
	ETR	3.6 ± 0.54	3.2 ± 0.62	3.8 ± 0.85	3.2 ± 0.43	ns	ns	ns
60	F_v_/F_m_	0.81 ± 0.003	0.81 ± 0.008	0.81 ± 0.007	0.80 ± 0.011	ns	**	ns
	NPQ	0.15 ± 0.005 *c*	0.17 ± 0.008 *b*	0.18 ± 0.008 b	0.22 ± 0.011 a	***	***	*
	ETR	3.0 ± 0.19	2.9 ± 0.05	3.0 ± 0.13	2.4 ± 0.39	*	**	ns
	RGR	0.03 ± 0.006	0.04 ± 0.002	0.03 ± 0.005	0.03 ± 0.009	ns	ns	ns

## Data Availability

The data presented in this study are available on request from the corresponding author.

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
