# Peer review of "The Role of Aquaporin Overexpression in the Modulation of Transcription of Heavy Metal Transporters under Cadmium Treatment in Poplar"

_plants, 2020, doi:10.3390/plants10010054_

Round 1

Reviewer 1 Report

No comments

Author Response

.

Reviewer 2 Report

To the authors,

the manuscript ID: plants-969467 reports the modulation of transcription of heavy metal transporter under Cd treatment in two lines of Populus alba (L.) “Villafranca” clone wt and over-expressing the aquaporin (aqua1). Generally the manuscript is clearly written and well documented in all parts of it. However, I suggest to consider and clarify a few points reporter below:

line 12: …an aquaporin (GenBank… should be written… an aquaporin (aqua1, GenBank…

line 25: I have some doubts regarding keywords, for example right keywordis metal transport or metal transporters?

line 166: Please check the value 14.8%, maybe is 13.3%.

line 183: I suggest to insert in Table 1 the meaning of Fv/Fm, NPQ and ETR as you have done for RGR. Moreover, check day, parameter… line, please.

Line 192: biological replicates + SE… should be… biological replicates ± SE. If it is possible, increase the font of the letters in Figure 1.

Line 199: two genes (Potri.007G050600 and Potri.007G050700) are omologue to one gene (AtNRAMP3). Check, please.

Line 245: Figure 3. Why did you change the gene order in the list respect to Figure 2? Is it possible to fix them?  

Line 253: biological replicates + SE… should be… biological replicates ± SE. If it is possible, increase the font of the letters in Figure 4.

Line 273: I have some doubts regarding “different functionality” probability is better “differential functionality”.

Line 277: to uptake Cd… I suggest… to translocate Cd.

Line 297: There is a recent review regarding the absorption of heavy metals in plants De Caroli et al., 2020, Plants 9, 482, doi: 10.3390 / plants9040482 that you can include here.

Line 299: vacuoles membrane… should be… vacuoles tonoplast.

In the supplementary Figure you should check if you have written biological replicates ± SE.

Author Response

Reviewer: 2

Comment#1: Line 12: …an aquaporin (GenBank… should be written… an aquaporin (aqua1, GenBank…

Answer: Thank you for your suggestion. We introduced aqua1 as suggested in the abstract section.

Comment#2: Line 25: I have some doubts regarding keywords, for example right keywords metal transport or metal transporters?

Answer: We changed the keywords accordingly.

Comment#3: Line 166: Please check the value 14.8%, maybe is 13.3%.

Answer: Thank you for your suggestion. We changed the value with the correct percentage (19.6%).

Comment#4: line 183: I suggest to insert in Table 1 the meaning of Fv/Fm, NPQ and ETR as you have done for RGR. Moreover, check day, parameter… line, please.

Answer: The legend of Table 1 has been improved as follows:

“Table 1. Photosystem II efficiency parameters at 1, 7, and 60 d of Cd treatment and relative growth rate (RGR, g day−1), Fv/Fm (maximum quantum efficiency of photosystem II), NPQ (non-photochemical quenching), ETR (electron transport rate) ……”

Moreover, we checked the day, parameter, letters and lines as indicated by the Reviewer.

Comment#5: Line 192: biological replicates + SE… should be… biological replicates ± SE. If it is possible, increase the font of the letters in Figure 1.

Answer: The legend has been improved and the letters in the graph increased.

Comment#6. Line 199: two genes (Potri.007G050600 and Potri.007G050700) are omologue to one gene (AtNRAMP3). Check, please.

Answer: Thank you for the comment. We have further checked this information and these two genes are both homologues of AtNRAMP3.

Comment#7. Line 245: Figure 3. Why did you change the gene order in the list respect to Figure 2? Is it possible to fix them?

Answer: In the Figure 2 the heat map describes the relationship among gene transcription levels in wt and aqua1 plants at 1, 7, and 60 d in control and treated conditions. In Figure S2 the heat map reports the comparison between wt and aqua1 gene transcription levels in control and treated plants at 1 day. In these heat maps the gene order is assigned on the basis of a correlation matrix and therefore cannot be modified since gives information about the genes that have similar transcription patterns (e.g., NRAMP3.1 and ABCC13 or NRAMP2 and ABCC9).

Comment#8: Line 253: biological replicates + SE… should be… biological replicates ± SE. If it is possible, increase the font of the letters in Figure 4.

Answer: The legend of Figures  have been improved and the font of the letters in the graph increased.

Comment#9: Line 273: I have some doubts regarding “different functionality” probability is better “differential functionality”.

Answer: We agree with the suggested change and we have done it.

Comment#10: Line 277… to uptake Cd… I suggest… to translocate Cd.

Answer: We agree with the suggested change and we have done it.

Comment#11: Line 297: There is a recent review regarding the absorption of heavy metals in plants De Caroli et al., 2020, Plants 9, 482, doi: 10.3390 / plants9040482 that you can include here.

Answer: Thank you for your suggestion. The reference has been added along the manuscript (References n. 50, De Caroli, M.; Furini, A; DalCorso, G; Rojas M; Di Sansebastiano G.P. Endomembrane reorganization induced by heavy metals. Plants 2020, 9, 482; doi:10.3390/plants9040482.).

Comment#12: Line 299: vacuoles membrane… should be… vacuoles tonoplast.

Answer: We agree with the suggested change and we have done it.

Comment#13: In the supplementary Figure you should check if you have written biological replicates ± SE.

Answer: All the captions and figures in supplementary materials have been improved.

Reviewer 3 Report

the paper takes an interesting approach to understanding how Cd enters and is transported within a woody plant

there are several sticky notes where suggestions for changes are made

there are several suggested modifications for changes within the abstract

a summary figure or even a table where the different transporters are described would be useful to a reader (save multiple note taking)   

and because there was full nutrition from Hoaglands its possible that there was interactions between  Cd and other divalent metals (Fe, Mn, Cu, Zn) that are also transported with some of the transporters

I looked for a concluding statement about whether a GMO plant might be useful in clean up    

it seems that there is a subtle role of aquaporins in uptake/ transport of Cd --  can this be strengthened in the conclusion?

what would the neutral complex of Cd be for these aquaporins?

also it would be valuable from the scientific view to have a comparison with the findings from the Zn aquaporin studies 

Author Response

Reviewer: 3

Comment#1: there are several sticky notes where suggestions for changes are made

Comment#2: there are several suggested modifications for changes within the abstract

Answer: the suggested changes reported into the sticky notes in the manuscript uploaded by Reviewer have been done.

In particular:

  • Line 10: The word “determine” rather than “deepen” has been used.
  • Line 12: The gene has been defined.
  • Line 12: The growth medium has been added.
  • Line 14: was the dose of Cd one time or repeated over the total study time? The dose of Cd was renewed every 5 d. We added this information within the Abstract.
  • Line 16: what happens to the wt? can you clarify? More information has been added concerning wt plants.
  • Line 20: line? do you mean wt v   aquaporin mutant. The sentence “between lines” has been removed.
  • Line 44: are all these genes transcribed in the root, are they in the same tissues of the root, root hair v epidermis v cortex etc. The information about gene expression has been removed from the introduction section and reported in materials and methods and results sections.
  • Line 55: translocation or storage for protection? The word has been changed to “storage for protection”.
  • Line 88: please could you explain here in what tissues aqua1 is expressed root only? stele etc important for the reader to know. so i went to your reference and learned that it could be roots and leaves'. . The AQUA1:GFP line over-expressing aqua1 revealed a strong localization of this gene in roots and leaf guard cells on in vitro plantlets [42].
  • Line 90: please explain these terms. The extended word “Wood Plant Medium” has been added to the manuscript.
  • Line 96: would not the clay sorb the metal ions? No, expanded pebble clay is an inert substrate.
  • Line 113: I may be wrong but i thought Fv/Fm was the efficiency and xPSII the quantum max yield? The correct definitions have been reported in the manuscript.
  • Line 120: the word “further” has been removed.
  • Line 139: the verb “ground” has been modified.
  • Line 171: We changed the sentence to: “…while it was below the detection limit within the shoot”.
  • Line 173: what is the stat analysis here are these the same or different? The values are not different, we changed “vs.” to “and”.
  • Line 176: can you define medial? i presume it is the mid section of the shoot? may be give cm lengths of all three sections. what about basal shown in figure? Leaves were separated in apical (1 ≤ LPI ≤ 6), medial (7 ≤ LPI ≤ 18), and basal (LPI > 18) on the basis of leaf plastochron index. This information is reported at lines 121-122. The description of Cd trend in basal leaves has been added.
  • Line 200: think your abstract and title should reflect that the study is more than making the aqua 1 mutant and Cd uptake you actually data mined for all of these transport genes. If so modeling this complex former to compare Cd Cu Fe and Zn might be useful. Could you indicate too what is the form of the transported metal conjugate is this nicotianamine? Thank you for your suggestion. Several complex forms and conjugates are reported in literature, one of them is nicotianamine. Our manuscript did not focus on this aspect that could be an interesting starting point for future works.
  • Line 209: for what plant type? not clear without looking at figure. We improved the description of the heat map.
  • Line 212: its transcript accumulation that you actually measured. We modified the sentence accordingly.
  • Line 237: the word “fold” has been correct.
  • Line 257: use ug/ml to be consistent with how you describe the dose of Cd. μg g-1 has been used in the text instead of ppm.
  • Line 284: a diagram with all these transporters would be very useful. Table 1 has been added to the manuscript.

Comment#3: a summary figure or even a table where the different transporters are described would be useful to a reader (save multiple note taking)

Answer: As suggested, we added Table 1 in Materials and Methods with the main information about the different transporters investigated in the manuscript.

Comment#4: because there was full nutrition from Hoaglands its possible that there was interactions between Cd and other divalent metals (Fe, Mn, Cu, Zn) that are also transported with some of the transporters.

Answer: We cannot exclude possible interactions of divalent metals with Cd. Indeed, we reported Mn and Zn concentrations (the two divalent cations most transported in combination with Cd) within Fig.2 and Supplementary material Fig.S1 to evaluate possible interactions between these metals. As discussed in the manuscript, Mn was lower in both treated lines.

Comment#5: I looked for a concluding statement about whether a GMO plant might be useful in clean up

Answer: Thank you for the suggestion. We added the following sentence:

“This work further confirmed the importance of exploring the use of transgenic plants with improved heavy metal tolerance for promoting sustainable phytoremediation practices [68].”

Comment#6: it seems that there is a subtle role of aquaporins in uptake/ transport of Cd -- can this be strengthened in the conclusion?

Answer: Thank you for the suggestion, we modified the conclusion in order to strengthen this concept.

Comment#7: what would the neutral complex of Cd be for these aquaporins?

Answer: The function of aquaporins is facilitating the movement of water across membranes. However, some evidences report that cations could be transported by aquaporins as well. For example, Li et al. (2016, Plant Cell Physiology, 57(1), 4–13) demonstrated that members of the nodulin 26-like intrinsic protein family of plant aquaporins are involved in As(III) uptake by roots in Arabidopsis thaliana and Oryza sativa plants. A study on Allium cepa indicates that heavy metals, such as Cd, gate aquaporins in the membranes of epidermal cells (Przedpelska-Wasowicz et al., 2011, Protoplasma, 248, 663–671). Recently, it has been demonstrated that the grapevine NIP2;1 aquaporin is also a silicon channel (Noronha et al., 2020, Journal of Experimental Botany, 294). Therefore, even if heavy metals such as Cd are usually chelated by phytochelatins, literature supports that aquaporins do not transport metals-conjugates (Maurel et al., 2015, Physiological Reviews, 95(4), 1321–1358).

Comment#8: also it would be valuable from the scientific view to have a comparison with the findings from the Zn aquaporin studies

Answer:

Thank you for the suggestion. We added the following lines:

315-317. Ariani et al. [42] reported that aqua1 plants under Zn stress reported no difference in Zn concentration between line while as in our experiment the metal stress affected mineral nutrient uptake.

327-329. Zn stress applied to aqua1 plants affected the Fv/Fm only in the apical leaves after 35 days of treatment showing a more stressed photosynthetic system in comparison to the Cd stress applied in the current research [42].

Reviewer 4 Report

The manuscript Plants-969467 by Neri et al. entitled “The role of aquaporin overexpression in the modulation of transcription of heavy metal transporters under cadmium treatment in poplar” studies the response to cadmium exposure of transgenic poplar plants over-expressing an aquaporin compared to wild type, analyzing at transcriptional level some heavy metal transporter genes.  

Indeed, not enough information about this topic is present in literature, making the paper interesting but, in the complex, it is not completely investigated.

The introduction should be further developed and strengthened by adding updated references both for heavy metal transporter genes and for aquaporin proteins and their correlation during heavy metal exposure, especially cadmium.

In the Material and Methods section, the description is sometimes too much detailed and the information are not useful to make clear the used test conditions.

Results section is not easily readable; many described results are in the supplemental part.

Table 1 is badly formatted and it is not explained the significative increase of NPQ at 60 days in overexpressed plants. In figure 1, the control plants, strangely, show higher content of Cd compared to plants grown in presence of Cd even if not significative. In general, the figures are filled with too many informations and it is hardly possible to interpret the results. The description of the Figure 2 is approximate (lines 209-211). The heatmap in Figure 3 is not well-described (lines 210-214).

It would be interesting, in addition, to support the conclusion concerning the improvement in Cd transport of aqua1 plants, to measure the expression levels of γ-glutamylcysteine synthetase (γ-GCs), glutathione synthetase (ECGs) and phytochelatin synthase (PCs) genes in plant leaves.

The discussion of the results is too theoretical when referred to the presented conclusions.

The English used in the paper should be accurately revised.

Author Response

Reviewer #4

Comment#1: Indeed, not enough information about this topic is present in literature, making the paper interesting but, in the complex, it is not completely investigated.

Answer: We agree with this comment. The novelty of this work is indeed a point of strength but do not allow an easy discussion of results. However, we further improved introduction and discussion as also remarked by other Reviewers.

Comment#2: The introduction should be further developed and strengthened by adding updated references both for heavy metal transporter genes and for aquaporin proteins and their correlation during heavy metal exposure, especially cadmium.

Answer: Thank you for the suggestion. We added lines 31-32, lines 57-59, and lines 63-64 with updated references in the introduction.

Comment#3: In the Material and Methods section, the description is sometimes too much detailed and the information are not useful to make clear the used test conditions.

Answer: Thank you for the suggestion, we shortened the Materials and Methods section as follows:

Line 91 conditions in Magenta vessels

Line 99 used in the experiment. Plantlets maintained

Line 100 ….from 100% to 65–70% to adapt plants to the growth chamber conditions

Line 101 randomly dived in two groups and

Line 106 of each plant parts or the whole plant

Line 107 Hoagland’s solution

Line 125 To deepen the effect of Cd treatment in the new formed leaves at

Line 129 filtered, opportunely diluted with ultra-pure water and then

Line 144 in sterile mortars. Then,

Comment#4: Results section is not easily readable; many described results are in the supplemental part.

Answer: Thank you for the suggestion, we moved into the manuscript results about Mn concentration and gene expressions at 1, 7, and 60 days of Cd exposure.

Comment#5: Table 1 is badly formatted and it is not explained the significative increase of NPQ at 60 days in overexpressed plants.

Answer: Thank you for the suggestions. We improved the Table 1. Moreover, in the discussion section we explained the significative increase of NPQ at 60 days in overexpressed plants.

Lines 316-321: “After 60 d of Cd exposure, although wt and aqua1 leaves did not show any symptom of toxicity in terms of chlorosis, necrosis, and Fv/Fm parameter, an indication of plant response to Cd stress has been provided by NPQ increase. The NPQ is a key protective process for thermally dissipating the excess of light energy that plants employ to prevent the over reduction of PSII. A noticeable difference in this parameter was observed between aqua1 and wt plants at 60 d suggesting that the whole-chain electron transport was more affected by Cd exposure in aqua1 leaves.”

Comment#6: In figure 1, the control plants, strangely, show higher content of Cd compared to plants grown in presence of Cd even if not significative. In general, the figures are filled with too many informations and it is hardly possible to interpret the results.

Answer: Thank you for the suggestions. We improved the quality of Fig.1, resizing also the range of y-axis. Data of Cd are, in fact, below 1 µg g-1 DW.

Comment#7: The description of the Figure 2 is approximate (lines 209-211).

Answer: We improved the description as suggested by the Reviewer.

Comment#8: The heatmap in Figure 3 is not well-described (lines 210-214).

Answer: We improved the description as suggested by the Reviewer.

Comment#9: It would be interesting, in addition, to support the conclusion concerning the improvement in Cd transport of aqua1 plants, to measure the expression levels of γ-glutamylcysteine synthetase (γ-GCs), glutathione synthetase (ECGs) and phytochelatin synthase (PCs) genes in plant leaves.

Answer: Thank you for the suggestion. It will be an interesting starting point of investigation for future works.

Comment#10: The discussion of the results is too theoretical when referred to the presented conclusions.

Answer: We improved the conclusions as also suggested by the Reviewer 3. Moreover, we also strengthened the discussion.

Comment#11: The English used in the paper should be accurately revised.

Answer: Thank you for your suggestion, we checked English language and style.

Round 2

Reviewer 3 Report

the authors have paid attention to previous comments making several changes

i feel the science is OK - the statements about the findings are justified although the role of the aquaporin remains elusive based on the findings in this paper it remains factual providing details about location and extents of changes in expression of different transport genes this is because even after 60 d of exposure there is very little impact of Cd treatments in this resistant poplar line maybe they should have done studies with a sensitive line but that is not the case

however the paper for me with english as my first language still has many grammatical problems as suggested by the many sticky notes

and there are still flaws in the writing word suggestions etc are made again(see the attachment).

Author Response

Dear Rewiever,

thanks for your indication. We made all the suggested comments and the revised manuscript with changes marked in red has been uploaded.

We only did not modify the following sentences:

  • Line 93: we left in vitro plantlets instead change to seedlings. In fact, we micropropagated the plants starting from cuttings
  • Line 238: Reviewer sentence: Could you add to the heat map Fig. 3 the results for gene aqua1?

Answer: As reported by Ariani et al. [42] the aqua1 line over-expressing the aquaporin aqua1 69 times more than the wt plants. Transgenic lines carrying an AQUA1:HA construct under the 35S promoter, so in the heat map this will be appear as constant up regulation in all organs of aqua1 plants.

Best regards,

A. Andreucci

Reviewer 4 Report

The manuscript has been improved as requested, since the authors answered to all the questions presented. In particular, the results and discussion are much clearer in respect to the previous version.

Author Response

Thanls for your helpuf comments.

Best regards,

A. Andreucci